# Clinical and Epidemiological Characteristics of Mpox in HIV-Infected and Uninfected Men Who Have Sex with Men: A Retrospective Study in Lisbon

**DOI:** 10.3390/v16020225

**Published:** 2024-01-31

**Authors:** Diogo de Sousa, Yuliya Volovetska, Daniel Nunes, Carlos Lemos, João Borges-Costa, Paulo Filipe

**Affiliations:** 1Dermatology Department, Centro Hospitalar Universitário Lisboa Norte, 1349-008 Lisbon, Portugal; 2Clinical Pathology Department, Centro Hospitalar Universitário Lisboa Norte, 1349-008 Lisbon, Portugal; 3Dermatology University Clinic, Faculty of Medicine, University of Lisbon, 1169-050 Lisbon, Portugal; 4Instituto de Higiene e Medicina Tropical, Nova University of Lisbon, 1349-008 Lisbon, Portugal; 5Dermatology Research Unit, Instituto de Medicina Molecular, University of Lisbon, 1169-050 Lisbon, Portugal

**Keywords:** Mpox, HIV, MSM, epidemiology

## Abstract

The resurgence of Mpox, predominantly among men who have sex with men (MSM), has prompted an analysis of its clinical manifestations and epidemiological patterns, particularly in individuals living with human immunodeficiency virus (HIV). This retrospective study aims to delineate and compare the clinical presentations and epidemiological characteristics of Mpox among HIV-positive and HIV-negative individuals. A total of 58 confirmed Mpox cases from a dermatology department in Lisbon were analyzed, focusing on mucocutaneous manifestations, systemic symptoms, and laboratory parameters. Our findings indicate no significant difference in disease severity and presentation between HIV-positive and HIV-negative groups, suggesting that HIV status may not be a determinant of Mpox severity, but rather an indicator of increased sexual risk behaviors, a recognized risk factor for Mpox transmission.

## 1. Introduction

The recent resurgence of Mpox, a zoonotic disease caused by the Mpox virus belonging to the Orthopoxvirus genus in the Poxviridae family, has raised significant public health concerns globally [1,2]. Notably, this resurgence is marked by a deviation from the traditionally endemic regions in Central and West Africa to non-endemic countries, affecting new demographic groups, particularly men who have sex with men (MSM), a considerable proportion of whom are living with human immunodeficiency virus (HIV) [3]. The intersection of Mpox and HIV presents a unique and complex challenge, underscoring the necessity to understand the interplay between these two conditions for the purpose of developing effective clinical and public health strategies.

Historically confined to specific African regions, the disease’s spread to non-endemic areas indicates a shift in its epidemiological pattern, necessitating global vigilance and a reevaluation of its clinical and epidemiological characteristics [4,5,6]. The clinical manifestation of Mpox typically involves a prodrome phase characterized by fever, intense headache, lymphadenopathy, myalgia, and asthenia, as well as a typical rash that evolves through various stages before resolving [3,7]. While generally self-limiting, with symptoms lasting from 2 to 4 weeks, severe cases, particularly in immunocompromised individuals, can lead to severe organ complications, with dermatological, respiratory, and secondary bacterial infection being the more commonly reported [8,9].

HIV infection compromises the immune system, rendering individuals more susceptible to a range of opportunistic infections and affecting the epidemiology, clinical manifestation, and treatment outcomes of concurrent diseases. The interaction between HIV and Mpox, therefore, represents a critical area of investigation given the potential for altered disease severity and progression in co-infected individuals. Advanced, uncontrolled HIV infection has been linked to a higher need for hospitalization, prolonged disease courses, development of complications, or even death attributed to Mpox [8].

Set against this backdrop, the current retrospective study aims to describe and compare the clinical presentations and epidemiological characteristics of Mpox among individuals with and without HIV infection.

## 2. Materials and Methods

### 2.1. Participant Selection

Participants were selected from the pool of individuals diagnosed with Mpox who presented at the Dermatology Department of Centro Hospitalar Universitário Lisboa Norte, Lisbon, Portugal, between January and December 2022. A total of 58 patients were included, categorized into two groups based on their documented HIV status: HIV-positive and HIV-negative. Inclusion criteria consisted of a confirmed diagnosis of Mpox and age above 18 years. The diagnosis was confirmed with swab samples from the skin or mucosa lesions (suspected lesions at the clinician’s discretion). Laboratory diagnosis was performed at the Portuguese National Sexually Transmitted Infections (STI) Reference Centre with PCR in real time to detect the orthopoxvirus genus gene rpo18, followed by Sanger sequencing of PCR products from the lesions, as previously described [10]. Patients were excluded if they had incomplete medical records or if they declined consent for their data to be used in research. Ethical approval for the study was obtained from the hospital’s Institutional Review Board, and all procedures were conducted in accordance with the Declaration of Helsinki.

### 2.2. Data Collection

Clinical data were extracted retrospectively from electronic medical records. The data collected included demographic information (age, sex, MSM status), clinical presentation of Mpox (type and distribution of lesions, presence of systemic symptoms), and HIV status (if unknown). The laboratory parameters gathered included complete blood count, liver function tests, and C-reactive protein levels.

### 2.3. Statistical Analysis

Statistical analyses were performed using SPSS software (version 27.0.1). Descriptive statistics were used to summarize demographic and clinical characteristics. The *t*-test was employed for continuous variables to compare means between the HIV-positive and HIV-negative groups, while Fisher’s exact test was used for categorical variables to compare proportions. A *p*-value of less than 0.05 was considered statistically significant.

## 3. Results

### 3.1. Population

All 58 patients were male, predominantly MSM (88%) (Table 1). From the included 25 patients with HIV, 88% (*n* = 22) were on highly active antiretroviral therapy (HAART) and reported stable and virologically effective antiretroviral treatment. Most patients had a previous sexually transmitted infection (STI) diagnosis, 51.5% (*n* = 17) and 52.0% (*n* = 13), in the HIV-negative and HIV-positive groups, respectively. It total, 7 patients had received a smallpox vaccination: 2 (6.1%) in the HIV-negative group and 5 (20%) in the HIV-positive group.

### 3.2. Clinical Presentation

All 58 patients presented with mucosal and/or cutaneous lesions characteristic of Mpox. Skin or mucosal findings were more prevalent in the genital area (60.3%); followed by the perianal region (50.0%); and, lastly, the perioral region (25.9%) (Table 2). Clinical presentation at diagnosis did not show a significant difference in prevalence between the two groups.

Constitutional symptoms such as lymph node enlargement (*n* = 38; 65.6%), myalgias/arthralgias (*n* = 32; 55.2%), fever (*n* = 31; 53.4%), and headaches (*n* = 30; 51.7%) were prevalent among the cohort (Table 2). The frequency of these symptoms did not significantly differ between the HIV-positive and HIV-negative individuals.

Bacterial cutaneous infection of the lesions was the only complication reported, with three patients in each group: 9.1% in the HIV-negative group, and 12.0% in the HIV-positive group. No patient required antiviral treatment directed at Mpox.

### 3.3. Laboratory Findings

Of the 58 patients, 32.8% (*n*= 19) of patients had another concomitant STI diagnosis upon screening: 8 in the HIV-positive group (32.0%); and 11 in the HIV-negative group (33.3%). In total, 15 cases of primary/secondary syphilis were diagnosed, 2 *Mycoplasma genitalium*, 1 *Neisseria gonorrhoeae*, and 1 *Chlamydia trachomatis*. Laboratory evaluations, including red cell count, white cell count, renal function, liver function, and c-reactive protein, were similar in both groups, with no significant differences found (Table 3).

## 4. Discussion

Our study’s primary finding is the lack of significant differences in the clinical severity and laboratory parameters of Mpox between HIV-positive and HIV-negative individuals. This suggests that, while HIV-positive individuals are more likely to contract Mpox, likely due to behavioral factors associated with increased risk, their immune status does not significantly alter the course of the disease [3]. Notably, most HIV-positive patients included in our analyses were on HAART (88%). This can explain the absence of differences, as previous works have shown that people living with HIV (PLWH) with high CD4 cell counts (>350 cells per mm^3^) mount a poxvirus-specific T-cell response that is similar to those without HIV infection [11]. Thus, we believe that our findings are only expandable to similar settings, where most PLWH are on HAART.

In contrast, PLWH with low CD4 counts (<100 cells per mm^3^) and high HIV viral loads (>log4 copies per mL) are at increased risk of severe forms of Mpox [9]. It is hypothesized that a substantial proportion of Mpox-virus-specific CD4 T cells might die or be impaired due to either complete or abortive HIV infection, leading some authors to advocate that a severe form of Mpox, with systemic involvement with disseminated and necrotizing lesions, could also be an AIDS-defining condition [9].

Clinical Mpox presentation was similar in both groups. All of our patients had skin or mucosal lesions, as evidenced by the referrals of patients seen with skin manifestations in the emergency department to our dermatology department. The lesion distribution in our sample was similar to previously described Mpox characteristics in this outbreak [3,6], with most lesions localized in areas of close contact during sexual intercourse. Our findings corroborate other published works where similarities in clinical presentations between the two groups, especially in terms of lesion distribution and systemic symptoms, are common [3,12].

The clinical outcomes in this case series were reassuring, as most cases were mild, with no need for hospital admission, similarly to previously published data where no significant differences were found in the proportion of PLWH who required reviewing in the hospital emergency department or inward admission [13]. Common systemic features which were encountered included lymphadenopathy, myalgia, fever, and headache, results that were similar in both groups and that are in line with the most common symptom triad described in Mpox patients besides skin/mucosal lesions: lymphadenopathy, fever, and asthenia/lethargy [3,6]. Thus, in similar high-income settings, there is no clear evidence that the risk factors or clinical course differ according to HIV status, except for a higher prevalence of rectal lesions at presentation among PLWH [14], which we did not find in our sample. Similarly to previous works comparing Mpox presentation in HIV patients, our findings corroborate that PLWH had similar clinical presentations to those without HIV, including indicators of more widespread disease, such as extragenital lesions and nondermatological symptoms [13,15].

Mpox-directed treatment was not initiated in any patient in our sample, namely, with tecovirimat, given the clinical course of the patients and the National Guidelines at the time of diagnosis. However, international guidelines state that treatment of Mpox should be considered among persons with HIV infection, taking into account disease severity, degree of immunosuppression, and vulnerable sites of infection (e.g., the genitals or anus) [16].

The high proportion of patients on HAART is a possible explanation for the lack of differences found in the clinical and laboratory courses of Mpox. In fact, a previous work by Kowalsi et al. found remarkably similar results: In a sample of 43 patients, all except one were on stable and virologically effective antiretroviral treatment (88.4% with HIV viral load < 50 copies/mL), and no significant clinical or laboratory differences or complication rates between patients with and without HIV infection were found [15]. The severity was also similar in both groups, as the disease was self-limited with no severe cases or deaths, in line with our results [15].

The higher prevalence of Mpox in MSM, particularly those with HIV, aligns with current outbreak reports from non-endemic countries. This demographic trend emphasizes the need for targeted public health interventions and education in these communities. The findings have several implications for public health practice and policy. Firstly, the lack of increased severity in PLWH on HAART is reassuring and suggests that current Mpox management strategies can be uniformly applied across HIV statuses. However, this should not lead to complacency, as all individuals, regardless of HIV status, should receive appropriate care and attention during Mpox outbreaks. Secondly, the prominence of Mpox in MSM and the association with sexual networks suggest that public health messaging and interventions need to be strategically directed. Safe sex practices, awareness campaigns, and even vaccination in high-risk groups are strategies to consider.

Our study has relevant limitations. The retrospective nature limits our ability to capture all relevant data comprehensively and introduces potential biases in patient selection and data recording. For instance, there is an underreporting bias for possible complications, as only more serious complications were reported to the physician; other common complications such as pain were not included in the registries. The sample size was relatively small and from a single center, which might limit the generalizability of the findings and reduce the statistical power of our findings. PLWH on HAART did not report CD4 counts or HIV viral loads, which might represent an HIV severity bias. Thus far, in high-income settings, there is no clear evidence that the risk factors or clinical courses differ according to HIV status.

## 5. Conclusions

In conclusion, our study provides valuable insights into the clinical and epidemiological characteristics of Mpox in PLWH. The lack of significant differences in clinical severity between the two groups is a crucial finding that informs clinical management and public health strategies. Combating stigma and discrimination and providing person-centered care have been at the core of the HIV response and are essential for appropriate clinical and public health services for all evolving pandemics, including Mpox. As Mpox continues to affect new populations, understanding its interaction with conditions like HIV is essential for effective control and prevention.

## Figures and Tables

**Table 1 viruses-16-00225-t001:** Study population.

	HIV−	HIV+
**N (%)**	33 (56.9)	25 (43.1)
**Age**	**M (SD)**	**M (SD)**
Years	35 (9)	37 (9)
**Sex**	***n* (%)**	***n* (%)**
Male	33 (100)	25 (100)
**Nationality**	***n* (%)**	***n* (%)**
Portugal	20 (60.6)	20 (80.0)
Other	13 (39.4)	5 (20.0)
**Sexual preference**	***n* (%)**	***n* (%)**
MSM	31 (93.9)	23 (92.0)
Heterosexual	2 (6.1)	2 (8.0)
**HAART**	-	***n* (%)**
	-	22 (88.0)
**Previous STI**	***n* (%)**	***n* (%)**
Yes	17 (51.5)	13 (52.0)
No/Unknown	16 (48.5)	12 (48.0)
**Vaccination against smallpox**	***n* (%)**	***n* (%)**
Yes	2 (6.1)	5 (20.0)
No	25 (69.7)	15 (60.0)
Unknown	6 (18.2)	5 (20.0)
**Number of sexual partners in the previous 3 months**	***n* (%)**	***n* (%)**
<10	29 (87.9)	20 (80.0)
>10	2 (6.1)	4 (16.0)
Unknown	2 (6.1)	1 (4.0)

HAART: highly active antiretroviral therapy. M: mean value. MSM: men who have sex with men. SD: standard deviation.

**Table 2 viruses-16-00225-t002:** Clinical data.

	HIV−	HIV+	*p*
**Genital lesions, *n* (%)**	**33 (100)**	**25 (100)**	0.300
Yes	18 (54.5)	17 (68.0)	
No	15 (45.5)	8 (32.0)	
**Perianal lesions, *n* (%)**	**33 (100)**	**25 (100)**	0.791
Yes	17 (51.5)	12 (48.0)	
No	16 (48.5)	13 (52.0)	
**Perioral lesions, *n* (%)**	**33 (100)**	**25 (100)**	0.778
Yes	9 (27.3)	6 (24.0)	
No	24 (72.7)	19 (76.0)	
**Lesions in other locations *n* (%)**	**33 (100)**	**25 (100)**	0.120
Yes	13 (39.4)	15 (60.0)	
No	20 (60.6)	10 (40.0)	
**Fever, *n* (%)**	**33 (100)**	**25 (100)**	0.847
Yes	18 (54.5)	13 (52.0)	
No	15 (45.5)	12 (48.0)	
**Myalgias/arthralgias, *n* (%)**	**33 (100)**	**25 (100)**	0.672
Yes	19 (57.6)	13 (52.0)	
No	14 (42.4)	12 (48.0)	
**Headache, *n* (%)**	**33 (100)**	**25 (100)**	0.971
Yes	17 (51.5)	13 (52.0)	
No	16 (48.5)	12 (48.0)	
**Adenopathies, *n* (%)**	**33 (100)**	**25 (100)**	0.144
Yes	19 (57.6)	19 (76.0)	
No	14 (42.4)	6 (24.0)	

**Table 3 viruses-16-00225-t003:** Laboratory data.

	HIV−		HIV+		*p*
	M	*n*	M	*n*	
**Hemoglobin (g/dL)**	14.2	23	14.8	19	0.119
**Leukocytes (10^9^/L** **)**	8.2	23	8.0	19	0.761
**Neutrophils (10^9^/L** **)**	5.2	23	4.8	19	0.536
**Eosinophils (10^9^/L** **)**	0.2	23	0.3	19	0.566
**Basophils (10^9^/L** **)**	0.1	23	0.1	19	0.845
**Lymphocytes (10^9^/L** **)**	2.6	24	3.5	18	0.530
**Monocytes (10^9^/L** **)**	1.3	24	1.4	18	0.918
**Platelets (10^9^/L** **)**	224.3	24	187.8	18	0.080
**Urea (mg/dL)**	26.9	25	26.2	17	0.763
**Creatinine (mg/dL)**	1.0	25	0.9	17	0.077
**Sodium (mmol/L)**	136.5	23	137.6	17	0.184
**Potassium (mmol/L)**	4.1	22	4.1	17	0.915
**ALT (U/L)**	35.8	24	48.1	16	0.273
**AST (U/L)**	26.4	24	36.4	16	0.142
**Bilirubin (total) (mg/dL)**	0.4	11	0.5	9	0.550
**LDH (U/L)**	191.4	11	228.6	8	0.226
**GGT (U/L)**	46.5	15	43.4	8	0.861
**Calcium (mg/dL)**	9.3	6	9	5	0.175
**Magnesium (mg/dL)**	4.3	6	2	7	0.363
**Creatine kinase (U/L)**	136.7	6	54	3	0.022
**CRP (mg/dL)**	4.0	25	3.5	16	0.621

ALT: alanine transaminase. AST: aspartate aminotransferase. CRP: C-reactive protein. GGT: gamma-glutamyl transferase. LDH: lactate dehydrogenase. M: mean value.

## Data Availability

Data are contained within the article.

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
