# Peer review of "Clinical and Epidemiological Characteristics of Mpox in HIV-Infected and Uninfected Men Who Have Sex with Men: A Retrospective Study in Lisbon"

_viruses, 2024, doi:10.3390/v16020225_

Round 1

Reviewer 1 Report

Comments and Suggestions for Authors

The manuscript titled "Clinical and Epidemiological Characteristics of Mpox in HIV-Infected and Uninfected Individuals: A Retrospective Study in Lisbon" by Diogo is a nice piece of work. The draft looks great but need minor revision before can be accepted for publication. My specific comments related to manuscript are as below

1) Minor writing editing or improvement

2) If possible increase the size of sample used for analysis

3) Information related to HIV patients means how long individuals were suffering form HIV infection as recently infected individual will behave OK and problems comes only after couple of years after infection. So this information is must for this study.

4) data provided in table suggested that HIV infection is recent, please clear this in manuscript, if HIV infection was old let say 5 years, than data in table will looks different. Please clear this

Comments on the Quality of English Language

Over all english is OK

Author Response

Dear reviewer,

We thank you for the kind comments. Please find the changes made to the manuscript below.

Best regards,

1) Minor writing editing or improvement

R: Corrections were made to improve writing and  correct minor errors.

2) If possible increase the size of sample used for analysis

R: The sample size used compromises all the patients that were eligible during the time frame analyzed. We are aware that an increase in the number of patients would allow for stronger conclusions to be drawn, however we are unable to increase the number of patients, as we are one medical center only.

3) Information related to HIV patients means how long individuals were suffering from HIV infection as recently infected individual will behave OK and problems comes only after couple of years after infection. So this information is must for this study.

R: All patients under HAART reported stable and virologically effective antiretroviral treatment, however, CD4 counts and HIV viral load where not available, as these are considered protected information by statal law. Nonetheless a statement was added referring reported effective antiretroviral treatment.

4) data provided in table suggested that HIV infection is recent, please clear this in manuscript, if HIV infection was old let say 5 years, than data in table will looks different. Please clear this

Patients in the HIV-positive group were mostly under HAART, with effective disease control, however duration of disease was not reported, and given the retrospective character of this study such data cannot be included in the analysis.  A reference in the study limitations was also made.

Reviewer 2 Report

Comments and Suggestions for Authors

De Sousa et al. are interested in the comparisons of clinical presentations and epidemiological characteristics of Mpox between HIV-positive and -negative individuals. The authors analyzed 58 Mpox cases from Lison, Portugal, and found no significant difference in disease severity and presentation between HIV-positive and -negative groups. Therefore, they concluded that HIV status may not be a determinant of Mpox severity. I have the following comments that the authors could consider.

1.     The title is misleading. The study mainly focused on the Mpox in HIV status of the MSM group (88%) rather than general HIV-infected and -uninfected individuals. Please revise it accordingly.

2.     Why are only 33 Mpox patient samples in the HIV-negative group?

3.     Please specify in more detail how the Mpox was confirmed (lines 60-61). At least, a relevant reference should be cited.

4.     Delete the “+” (line 84)

5.     Please unify “Mpox”, “mpox”, “monkeypox”, and “MPX” in the main text (lines: 134, 135, 142, 187).

6.     Please check the manuscript carefully to avoid grammatical errors (lines: 146, 154, 180).

7.     Please define “SD” in Table 1.

8.     Please define “M” in Table 3.

9.     A reference is needed to support this statement (lines 161-162).

10.   Discussion: Are there any similar studies performed previously? If so, the author should discuss them. If not, they could emphasize the novelty of the present study.

Comments on the Quality of English Language

Minor editing of English language required

Author Response

Dear reviewer,

We thank you for the kind comments. Please find the changes made to the manuscript below.

Best regards,

  1. The title is misleading. The study mainly focused on the Mpox in HIV status of the MSM group (88%) rather than general HIV-infected and -uninfected individuals. Please revise it accordingly.

R: The title was modifief to “Clinical and Epidemiological Characteristics of Mpox in HIV-Infected and Uninfected Men Who Have Sex with Men: A Retrospective Study in Lisbon” for a more accurate representation of the study.

  1. Why are only 33 Mpox patient samples in the HIV-negative group?

The number of Mpox patients in the HIV negative group can be explained by the disproportionally high  rates of Mpox in HIV positive groups, which is in line with other published samples (example: Tarín-Vicente EJ, Alemany A, Agud-Dios M, Ubals M, Suñer C, Antón A, et al. Clinical presentation and virological assessment of confirmed human monkeypox virus cases in Spain: a prospective observational cohort study. Lancet (London, England) [Internet]. Lancet; 2022 [cited 2024 Jan 6];400:661–9. Available from: https://pubmed.ncbi.nlm.nih.gov/35952705/)

  1. Please specify in more detail how the Mpox was confirmed (lines 60-61). At least, a relevant reference should be cited.

A description of the methods used for molecular diagnosis was added, with the relevant reference.

  1. 4. Delete the “+” (line 84)

Changed accordingly.

  1. Please unify “Mpox”, “mpox”, “monkeypox”, and “MPX” in the main text (lines: 134, 135, 142, 187).

Changed accordingly, Mpox was used to unify the different terms.

  1. Please check the manuscript carefully to avoid grammatical errors (lines: 146, 154, 180).

A full revision of the manuscript was made, including the noted errors.

  1. Please define “SD” in Table 1.

Definition added, as standard deviation.

  1. Please define “M” in Table 3.

Definition added, as Mean value.

  1. A reference is needed to support this statement (lines 161-162).

A reference was added to support the statement.

  1. 10. Discussion: Are there any similar studies performed previously? If so, the author should discuss them. If not, they could emphasize the novelty of the present study.

Previous published paper were added and discussed on the conclusions.

Reviewer 3 Report

Comments and Suggestions for Authors

This brief report describes a comparison of severity of Mpox infection in HIV+ and HIV- mostly MSM groups.  As the authors note, the major limitation of the study is that the group size is relatively small (58 total) and in a single location. The topic is significant and the information that the symptoms are similar in the (mostly on ART) HIV+ and HIV- groups is critically important for treatment considerations.  The data are straightforwardly presented and analyzed, although a stickler might object to the use of T-test without testing for normal distributions. The assertion that the MSM are more susceptible because of increased risk behaviors seems reasonable, but is speculative without an age matched non MSM control group, selected without knowledge of symptoms. The suggestion that Mpox increases are a warning indication that safer sex education is warranted, is an important conclusion of this report.

Comments on the Quality of English Language

 Minor comments are that "under HAART" might be replaced by "on HAART" throughout and that on page 4, line 128, "where" should be replaced with "were".

Author Response

Dear reviewer,

We thank you for the kind comments. Please find the changes made to the manuscript below.

Best regards,

This brief report describes a comparison of severity of Mpox infection in HIV+ and HIV- mostly MSM groups.  As the authors note, the major limitation of the study is that the group size is relatively small (58 total) and in a single location. The topic is significant and the information that the symptoms are similar in the (mostly on ART) HIV+ and HIV- groups is critically important for treatment considerations.  The data are straightforwardly presented and analyzed, although a stickler might object to the use of T-test without testing for normal distributions. The assertion that the MSM are more susceptible because of increased risk behaviors seems reasonable, but is speculative without an age matched non MSM control group, selected without knowledge of symptoms. The suggestion that Mpox increases are a warning indication that safer sex education is warranted, is an important conclusion of this report.

R: Thank you for your noteworthy comments. Regarding the assertion that MSM are more susceptible to Mpox infection is indeed based on observational studies, including ours. The fact that MSM are more susceptible to STI is noticeable through the literature. In the current outbreak Mpox is regarded as a STI, which can support the difference found in Mpox prevalence in MSM.

Minor comments are that "under HAART" might be replaced by "on HAART" throughout and that on page 4, line 128, "where" should be replaced with "were".

R: The noted errors were corrected throughout the manuscript.

Round 2

Reviewer 2 Report

Comments and Suggestions for Authors

The authors have adequately addressed my concerns in the revised manuscript. I have no more comments.